# Noisy Annotations in Segmentation

## COCO              Open Images

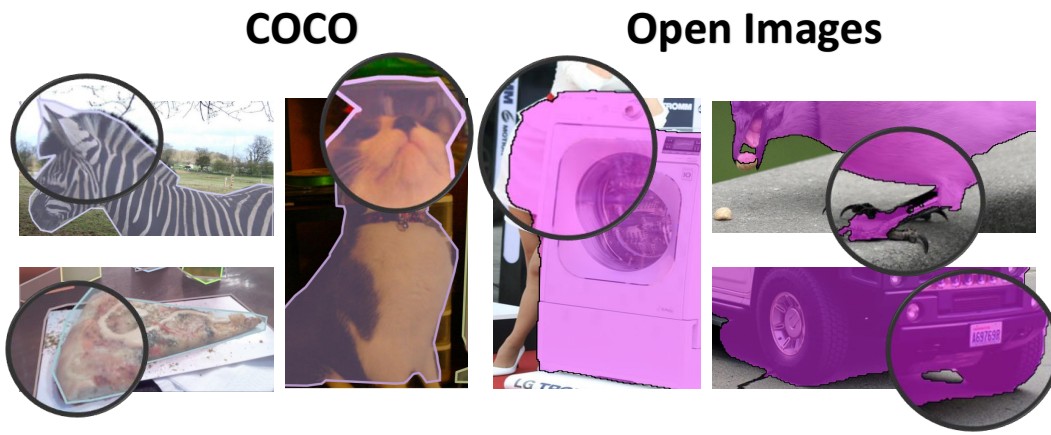

Figure 1: Annotation noise found in both manually labeled data and weakly annotated data. These errors include incomplete or over-extended masks, and ambiguous boundaries.

## Abstract

We propose four noise-augmented benchmarks—**COCO-N**, **CityScapes-N**, **VIPER-N** and the weak-annotation track **COCO-WAN**—that provide a unified test-bed for studying annotation noise in instance segmentation. A parametric engine stochastically perturbs mask boundaries, drifts spatial extents, flips categories and omits instances at three severity tiers, producing Monte-Carlo variants of any COCO-style corpus. Evaluating popular segmentation models such as Mask R-CNN, Mask2Former, YOLACT and SAM reveals up to 35 % drops in mask mAP under moderate noise, underscoring the limits of current learning-from-noisy-labels techniques when errors are spatial rather than purely categorical. All proposed **Benchmark-N** suite establishes a reproducible baseline for noise-aware segmentation and motivates future work on robust objectives, data-centric annotation pipelines and noise-adaptive architectures.

## 1 Introduction

Deep learning–driven instance segmentation underpins safety-critical applications ranging from autonomous driving to medical imaging. Its success hinges on *precise* pixel–level supervision, yet large, rapidly curated datasets inevitably contain erroneous masks. In echocardiography, for example, a modest 5% boundary error around the left-ventricular cavity can swing the ejection-fraction estimate from 45% to 39–50%, potentially tipping a diagnosis from borderline normal to pathological. Such high-stakes scenarios demand segmentation models that remain reliable when labels are imperfect.

Unfortunately, almost all noisy-label benchmarks—focuses on *class* noise for image classification. Spatial distortions, instance omissions and prompt-induced biases that plague instance segmentation

are far less explored, and there is no unified test bed for studying them at scale. Without realistic benchmarks, it is unclear how fragile current models are or which learning strategies truly help.

We close this gap with **Benchmark-N**, a suite of four noise-augmented datasets that inject empirically grounded spatial corruptions into both real (**COCO-N**, **CityScapes-N**) and synthetic (**VIPER-N**) data, plus a weak-annotation track (**COCO-WAN**) built with foundation-model prompts. A parametric generator produces controllable *boundary imprecision, spatial drift, category confusion* and *instance omission* at three severity tiers, enabling Monte-Carlo stress tests of any segmentation pipeline. Comprehensive experiments across Mask R-CNN, Mask2Former, YOLACT and SAM reveal sharp performance drops even under mild noise, exposing limitations of current learning-from-noise methods.

**Our contributions are:**

- A stochastic, task-agnostic noise model that synthesises diverse, realistic annotation errors for instance segmentation.
- Four publicly released benchmarks—**COCO-N**, **CityScapes-N**, **VIPER-N** and **COCO-WAN**—with reproducible "low/mid/high" noise presets.
- An extensive empirical study showing that popular CNN and transformer architectures lose up to ∼35% mAP under hard noise, underscoring the need for noise-aware training.

# 2   Related Work

**Noisy-label benchmarks.** Classification studies typically flip labels at random or via confusion matrices (CIFAR-N Wei et al. [2022], Clothing1M Xiao et al. [2015]); detection work jitters boxes or drops objects Mao et al. [2021], Ryoo et al. [2023]. In dense prediction, mask opening/closing Lu et al. [2014], Li et al. [2023] and class flips in medical data Nordström et al. [2022] leave object extent mostly intact, missing boundary jaggedness, spatial drift and omissions observed in practice.

**Weak or coarse labels.** Polygon-level Cityscapes-Coarse and Mapillary Vistas Cordts et al. [2016], click-based OpenImages Kuznetsova et al. [2020], and SAM-generated SA-1B Kirillov et al. [2023] support weak-supervised training but are not designed as robustness tests. Our **COCO-WAN** turns SAM masks—with controlled prompt noise—into such a benchmark.

**Learning with noisy labels (LNL).** Dense-task LNL adapts classification ideas: Adaptive Early-Learning Correction Liu et al. [2022], spatial Markov refinement Yao et al. [2023], and federated aggregation Wu et al. [2023]. Each uses bespoke or domain-specific corruptions, limiting comparability. Spatial noise thus remains largely un-benchmarked; our datasets provide the first multi-domain, reproducible test bed for boundary-level errors.

# 3   Annotation-Noise Generator

Accurate segmentation hinges on pixel–level agreement between an image and its ground-truth mask. In practice, annotation pipelines introduce *annotation noise*—any mismatch between the ideal (oracle) mask $M^*$ and the dataset mask $M$. We first catalogue common error modes, then formalise a stochastic generator that injects them with tunable severity.

## 3.1   Empirical Taxonomy of Annotation Errors

A manual sweep of COCO Lin et al. [2014], Cityscapes Cordts et al. [2016], OpenImages Kuznetsova et al. [2020] and LVIS Gupta et al. [2019] reveals four recurrent error families (illustrated in Fig. 1):

**Boundary Imprecision** — coarse or jagged outlines that over- or undershoot the true contour.

**Spatial Drift** — near-rigid shifts of an entire mask, typically caused by inattentive clicks or snapping heuristics.

**Category Confusion** — visually similar classes swapped (e.g. *bus→truck*), reflecting annotator ambiguity or taxonomy overlap.

**Instance Omission** — thin, occluded or low-contrast objects partially or fully omitted.

68  Automated polygon simplifiers, box-to-mask converters and prompt-based foundation models can
69  exacerbate these patterns by eroding fine structures or hallucinating plausible yet wrong regions.

## 3.2  Parametric Noise Model

71  Let $(M, c)$ denote a binary instance mask and its class label. We inject noise by composing five
72  independent perturbations; each perturbation is sampled *i.i.d.* per instance, so every invocation yields
73  a corrupted dataset.

74  *Approximation:* Simplify the polygon via Douglas–Peucker with tolerance $\varepsilon \sim \mathcal{N}^+(\mu_{\text{approx}}, \sigma_{\text{approx}})$.

75  *Localization:* Displace each vertex by $(\Delta x, \Delta y)$ where $\Delta x, \Delta y \sim \mathcal{N}(\mu_{\text{loc}}, \sigma_{\text{loc}})$ with random signs.

76  *Scale:* With equal probability, dilate or erode $M$ by a square kernel of size $K \sim$
77  $\max\{1, \lfloor \mathcal{N}(\mu_{\text{scale}}, \sigma_{\text{scale}}) \rfloor\}$. *Class Confusion:* With probability $p_{\text{cls}}$, replace $c$ by a sibling inside the
78  same super-category, following empirical confusion matrices Northcutt et al. [2021].

79  *Deletion:* With probability $p_{\text{del}}$, drop the instance altogether, mimicking missed objects.

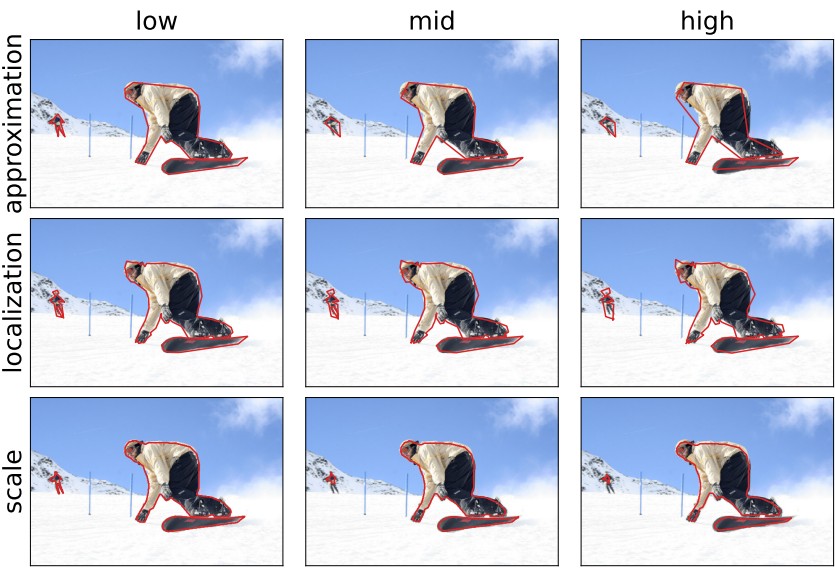

Figure 2: Illustrating the effects of the spatial noise with varying intensity.

## 3.3  Severity Presets and Reproducibility

81  Our open-source tool Benchmark-N suite[1] applies the above process to any COCO-style dataset.
82  Three presets—*Low*, *Mid*, *High*—scale $(\mu, \sigma, p_{\text{cls}}, p_{\text{del}})$ as detailed in Table 1. Because the generator
83  is purely stochastic, one can draw multiple corrupted variants, enabling Monte-Carlo robustness
84  studies instead of a single "clean vs. noisy" split.

85  This formulation cleanly decouples the *empirically grounded taxonomy* (Sec. 3.1) from the *synthetic*
86  *noise engine* (Sec. 3.2), providing a rigorous basis for analysing segmentation robustness under
87  realistic annotation imperfections.

88  All variables are sampled *i.i.d.* across instances, yielding a truly stochastic benchmark—unlike
89  previous works that commit to a single "clean vs. noisy" split Nordström et al. [2022], Lad and
90  Mueller [2023], Yao et al. [2023], Liu et al. [2022]. Three presets (*low/mid/high*) correspond to
91  increasing $(\mu, \sigma)$ pairs (Table 1, Appx.). Our public tool Benchmark-N applies these transformations
92  with a single command, enabling reproducible stress-tests of segmentation pipelines.

---

[1] https://anonymous.4open.science/r/noisy_labels-0C70/README.md

| Intensity | Low | Medium | High |
|---|---|---|---|
| $(\mu_{\text{approx}}, \sigma_{\text{approx}})$ | $(5, 2.5)$ | $(10, 2.5)$ | $(15, 10)$ |
| $(\mu_{\text{local}}, \sigma_{\text{local}})$ | $(2, 0.5)$ | $(3, 0.5)$ | $(4, 2)$ |
| $(\mu_{\text{scale}}, \sigma_{\text{scale}})$ | $(3, 1)$ | $(5, 1)$ | $(7, 4)$ |
| $p_{\text{class}}$ | 0.05 | 0.05 | 0.05 |
| $p_{\text{delete}}$ | 0.05 | 0.05 | 0.05 |

Table 1: Noise parameters used to produce the noisy annotations that compose **Benchmark-N**.

## 4 Benchmark

### 4.1 Synthetic Dataset: VIPER

In order to validate our noise model under perfectly labeled conditions, we turn to the VIPER dataset Richter et al. [2017], which is derived from the GTA V game engine. VIPER provides high-fidelity, pixel-accurate annotations for every object and region in the scene, making it a "clean" baseline for testing the pure effect of annotation noise.

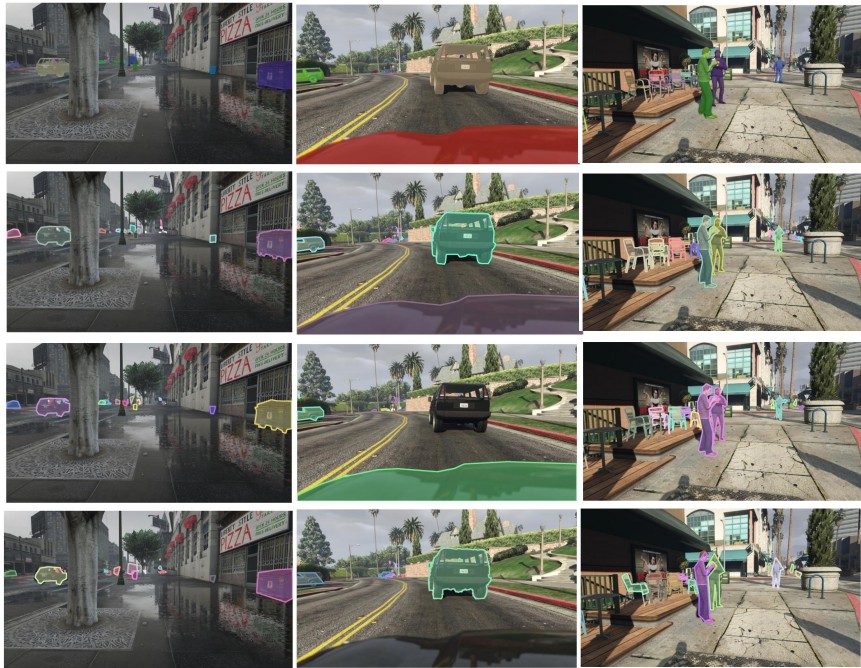

Figure 3: Examples from **VIPER-N** benchmark. Top row shows the clean annotations, second row the low noise regime, third present the midum annotation noise and last row the high annotation noise.

Because VIPER's segmentation maps are automatically rendered in a synthetic environment, the ground-truth annotations exhibit none of the spatial inaccuracies common in human-labeled datasets. This allows us to inject our prescribed noise types in a fully controlled way, without mixing in any preexisting labeling errors.

**Experimental Results** We train and evaluate the popular Mask R-CNN on VIPER-N and compare to the noise-free VIPER baseline. Figure 3 illustrates qualitative examples of clean vs. noisy labels, and Table 2 quantifies performance drops by model and noise level. Notably, even low-level spatial distortions can reduce precision significantly, confirming the sensitivity of modern architectures to subtle label corruptions.

Table 2: mAP on VIPER-N at four noise severities (higher is better)

| Noise | All | S | M | L |
|---|---|---|---|---|
| Clean | 15.8 | 6.0 | 44.3 | 60.6 |
| Low | 13.8 | 4.7 | 38.0 | 57.3 |
| Mid | 12.3 | 4.0 | 31.4 | 55.2 |
| High | 10.7 | 2.6 | 29.0 | 53.6 |

VIPER-N thus provides a controlled, synthetic test bed that highlights each model's vulnerabilities to annotation noise when all else—lighting, context, labeling scale—is held constant.

## 4.2 COCO-N & CityScapes-N

Finally, the noise integrated with the same noise strategies into widely used real-world datasets, producing **COCO-N** and **CityScapes-N**. Unlike VIPER, these datasets already contain minor human labeling errors, meaning our injected noise adds a further layer of realism. Below are the key steps and summary results.

We apply the exact same noise operations (§3.2) to each instance in COCO Lin et al. [2014] and Cityscapes Cordts et al. [2016] train splits. In line with VIPER-N, we create three tiers of severity (low, mid, high) by increasing the morphological kernel size, polygon simplification tolerance, and class confusion probabilities. Figure 7a illustrate the performance degradation on those, as well as LVIS Gupta et al. [2019] dataset, more details in the supp. materials.

**Results Across Popular Models.** Table 3 shows how varius models Mask R-CNN (R-50/R-101), Mask2Former (R-50/Swin), YOLACT fare on both **COCO-N** and **CityScapes-N** for all three noise tiers, as well as HTC Chen et al. [2019b] and SOLO Wang et al. [2020] for **COCO-N** Across the board, we see a notable dip in both standard mAP as well as boundary-focused metrics Cheng et al. [2021a] in supp materials. For **COCO-WAN**, we report fewer architectures, as full report will be released upon acceptance. Interestingly, transformer-based architectures (e.g., Swin in Mask2Former) appear slightly more robust to misaligned boundaries, but no model is immune to severe disruptions.

To assess the effect of label noise, we evaluate the performance of various instance segmentation models using our newly developed benchmark. We apply the various levels of noise, presenting **COCO-N** and **CityScapes-N**, providing insights into their robustness and adaptability. For more details about the models and datasets refer to the implantation details in the supplementary materials. Table 3 present the findings Mask-RCNN (M-RCNN) He et al. [2017], YOLACT Bolya et al. [2019], SOLO Wang et al. [2020], HTC Chen et al. [2019a] and Mask2Former (M2F) Cheng et al. [2021b]. **Clean** denote the performance of a model on the original annotations, where **Easy**, **Mid** and **Hard** correspond to the definition in Table 1. The reported numbers in the table represent mask mean average precision ($AP$) and boundary mask mean average precision ($AP^b$), respectively. More experiments involving LVIS dataset Gupta et al. [2019] and learning with noisy labels in sup. materials. All models trained and evaluated by standard training procedure[2]. We obtained additional experiments include cardiac unltrasound data in Appendix A, more evaluation metrics, models and datasets in Appendix E, and most notably, evaluate both **zero-shot** and **fine-tune** SAM Kirillov et al. [2023] on our proposed benchmark in Appendix F.

Our experiments demonstrate that label corruption leads to a degradation in model performance. Specifically, Mask R-CNN with a ResNet50 backbone retains approximately 80.6%, 71.7%, and 64.4% of its performance under Easy, Medium, and Hard noise conditions, respectively, on the **COCO-N** benchmark. The same model exhibits a more dramatic performance drop on the **CityScapes-N** benchmark, managing to retain only 72.8%, 60.9%, and only 45% under the corresponding noise levels. This trend is consistent across all tested models, suggesting that the impact is more crucial when less data is available, but might be easier to mitigate when using more data, even with the same portion of label noise.

This study demonstrates that all models are affected by labeling bias and exhibit diminished performance to varying extents, highlighting differing sensitivities to label noise. Notably, transformers

---

[2]openmmlab/mmdetection/model_zoo

Table 3: Evaluation Results of Instance Segmentation Models under Different Benchmarks, reporting mAP.

| Dataset | Model | Backbone | Clean | Easy | Mid | Hard |
|---|---|---|---|---|---|---|
| **COCO-N** | M-RCNN | R-50 | 34.6 | 27.9 | 24.8 | 22.3 |
| | YOLACT | | 28.5 | 26.4 | 23.3 | 20.8 |
| | SOLO | | 35.9 | 25.2 | 17.1 | 12.4 |
| | HTC | | 34.1 | - | 28.4 | 25.5 |
| | M2F | | 42.9 | 33.5 | 30.1 | 26.7 |
| | M-RCNN | R-101 | 36.2 | 28.8 | 31.8 | 23.7 |
| | M2F | Swin-S | 46.1 | 39.6 | 37.9 | 33.6 |
| **CityScapes-N** | M-RCNN | R-50 | 36.1 | 26.4 | 22.0 | 16.3 |
| | YOLACT | | 19.3 | 19.1 | 17.1 | 13.6 |
| | M-RCNN | R-101 | 37.0 | 33.7 | 30.7 | 27.0 |
| **COCO-WAN** | M-RCNN | R-50 | 34.6 | 32.8 | 24.4 | 21.6 |
| | Cascade M-RCNN | | 35.9 | 26.8 | 25.7 | 24.2 |
| | M2F | | 42.9 | 39.2 | 31.9 | 26.2 |
| | M2F | Swin-s | 46.1 | 42.9 | 34.4 | 28.4 |

display greater resilience, retaining 73% on the Hard benchmark, effectively mitigating the adverse effects of noisy labels compared to the convolution counterpart. This observation underscores the potential of using transformer-based architectures in scenarios where robustness to label noise is crucial. Our findings offer preliminary guidance for selecting or designing robust instance segmentation models in practical applications where encountering label noise is inevitable.

**Implications.** Given their critical role as mainstream benchmarks, **COCO-N** and **CityScapes-N** offer a practical measure of model reliability under imperfect labels. This can guide future research in developing noise-aware training strategies, data-cleaning pipelines, or architectures that gracefully handle label distortion. Our publicly released tool ensures that anyone can replicate these noisy benchmarks, tune the noise parameters, or adapt them to new datasets.

### 4.3 COCO-WAN (Weakly ANnotated)

Modern annotation pipelines commonly employ Vision Foundation Models (VFMs) Zhang et al. [2025] to reduce the dependence on fully manual labeling. While VFMs trained on large-scale data can produce high-quality masks, they often introduce systematic biases, since they overlook fine details. Due to the extend of tasks this models solves, for a specific context, they require some prompt that provides a task-specific context, as illustrated in fig. 4a. Specifically, we examine Segment Anything Model (SAM) Kirillov et al. [2023], prompting the model with either bounding-box, points, partial masks or text queries, incorporating noises based on the model and queries biases.

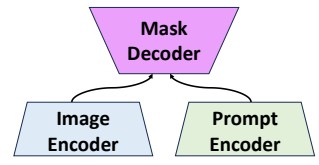

(a) prompt-based VFM. Points, boxes, and text guide the mask decoder.

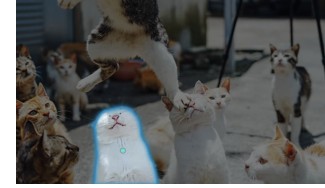

(b) Point Prompt

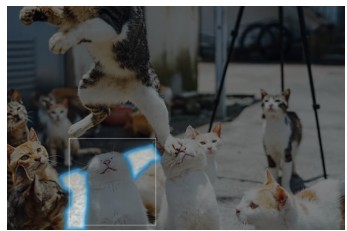

(c) Box Prompt

Figure 4: prompt-based VFM (left) and example SAM masks using different prompts (middle and right).

We have put into test three kinds of weak annotations as prompts, **Points**- one point per instance in the middle of the object mask. **Boxes**- the bounding box from the annotations, and **Text**- we

Table 4: COCO-WAN prompt quality (mAP, b-mAP Cheng et al. [2021a]) into Grounded-SAM.

| Prompt | Type | Clean | | Noisy | |
|---|---|---|---|---|---|
| | | AP | $AP^B$ | AP | $AP^B$ |
| Original labels | | 34.6 | 20.6 | - | - |
| Point | $\sim \mathcal{U}$ | 24.4 | 15.7 | 21.6 | 13.7 |
| Box | $+\mathcal{N}(0,2)$ | 32.8 | 19.7 | 25.3 | 15.9 |
| Text | {cls} | 22.0 | 14.1 | - | - |

Table 5: Mask2Former fine-tune vs. Gounded-SAM Ren et al. [2024] text prompted noise. Reporting $AP_m$ and $AP_b$ as mask and box mAP respectively.

| Model | Clean | | Noisy | |
|---|---|---|---|---|
| | $AP_m$ | $AP_b$ | $AP_m$ | $AP_b$ |
| R-50 | 42.9 | 45.7 | 26.2 | 24.1 |
| R-101 | 43.4 | 46.1 | 26.7 | 25.1 |
| Swin-S | 46.1 | 49.3 | 28.4 | 31.4 |
| Swin-B | 48.2 | 51.5 | 30.3 | 33.2 |

fed the class label from the annotations into Grounded-DINO, and used the boxes output as a box query, similar to Grounded-SAM Ren et al. [2024]. We examine the effect of noise on the prompts in Table 4, incorporate noise into the points, by randomly sample one point from the mask, and to the boxes by adding Gaussian noise ($\mathcal{N}(0,2)$) into one of the box corners.

In Table 5, we examine how a transformer backbone (Swin-S Liu et al. [2021]) impacts the Mask2Former Cheng et al. [2021b] model's robustness to noise. This noise degrades the models (on both mask and bounding box) by approximately $48\%$ in R-50 and $37.3\%$ in Swin-S. Although still notably affected by noise, this trend aligns with the results on **COCO-N** and **CityScapes-N** as reflected from Table 3.

Figure 4 illustrates how different prompt types can lead to varying degrees of segmentation noise, as for the given example bounding box captures the background instead of the actuall object, while a point is sufficient to produce high quality mask.

Qualitatively, SAM generally captures coarse object boundaries well, but Figure 5 shows how color and texture biases may cause missing or conflated parts, particularly in challenging scenes (e.g. without noticeable approximation errors). For instance, certain darker regions or closely colored objects can be merged or overlooked, signaling a lack of task oriented context. As a practical example, the middle image pair shows the pants and face of the standing person are not included in the mask due to the stark contrast in color from the light shirt. On the right image, we observe annotations with shape (stove-top) and instances of conflating potential objects (stove and cabinet) due to color biases. More qualitative results show in the supp. materials. In Figure 6 we see yet another example for auto-annotations excel in masks fidelity and even finding missing annotations, such as the portrait in the top-left pair, however, it commonly struggle with crowded annotations, as demonstrated in the bottom image, where the text was crowd orange and the mask include mostly the basket. this reflect the need to explore open vocabulary VFM that may overcome this annotation obstacle.

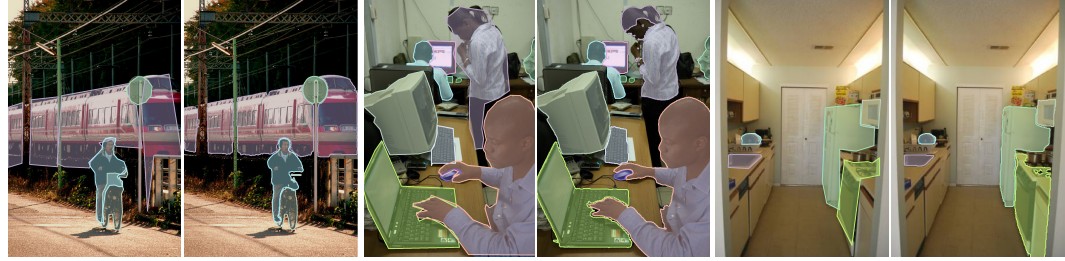

Figure 5: **Annotation quality** comparing COCO labels (left) and COCO-WAN labels using box queries (right)

This emphasizes the importance of developing more robust annotation strategies—both in prompt design and in subsequent label refinement—when relying on VFMs for real-world segmentation tasks.

## 4.4 Qualitatively Analysis

To evaluate how each noise independently affects model performance, we conduct an ablation using Mask R-CNN He et al. [2017] (ResNet-50 backbone) trained on the whole COCO with only one noise type active at various severity levels. Table 7 summarizes the quantitative impact on standard

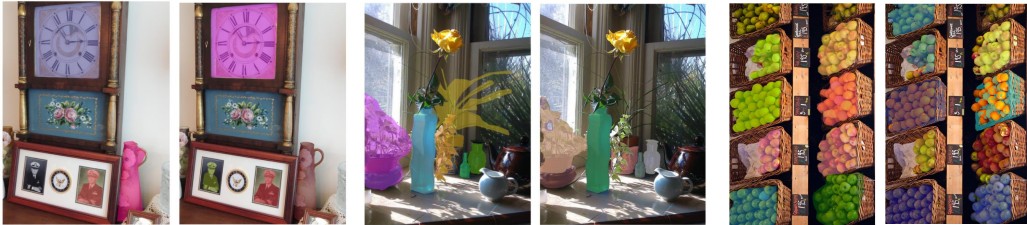

Figure 6: Compared annotations between COCO (left) and text prompt weak annotations (right).

metrics like mAP and boundary-level mAP (B-mAP) Cheng et al. [2021a]. Figure 11 visualizes performance declines for increasing noise severity.

*Scale Noise* (especially erosions) severely affects boundary fidelity, leading to the largest drop in performance, yet easy to fix by a pre-process morphological counter operation that bring the masks close to clean (e.g., opening or closing), thus, we chose to scale at random.

*Localization* and *Approximation Noise* subtly degrade object outlines, though moderate levels of displacement do not drastically lower global mAP.

*Class Confusion* chiefly impacts recognition accuracy; the reduced classification confidence leads to a measurable mAP drop, but less so on boundary metrics.

*Deletion* yields fewer total annotations, skews training and causes a performance loss.

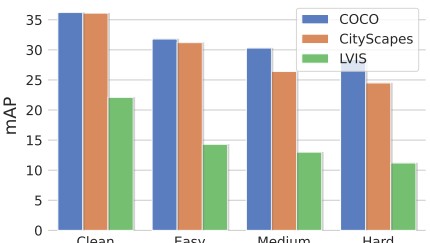

(a) Mask-RCNN performance on COCO, CityScapes and LVIS across three noise levels.

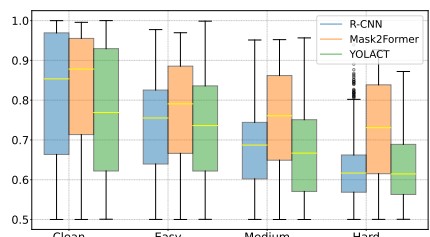

(b) Confidence scores (threshold $> 0.5$) of Mask-RCNN, Mask2Former and YOLACT under increasing noise.

Figure 7: Effect of annotation noise on segmentation quality (left) and prediction confidence (right).

Our experiments indicate that various architectures and backbones exhibit notable sensitivity to label noise, affecting both mask quality and prediction confidence. As shown in Figure 7b, higher noise levels correlate with reduced confidence scores, underscoring the vulnerability of model predictions to annotation accuracy. This effect is further illustrated in Figure 10, where increased noise leads to misclassification, causing the model to generate multiple conflicting predictions for a single instance.

## 5 Limitations

One limitation is that **Benchmark-N** suit targets four dominant error families (boundary imprecision, spatial drift, category confusion, instance omission). It does not yet cover multi-instance merge/split mistakes, or temporal label noise in videos. Future iterations should extend the taxonomy and validate it with larger human studies.

**COCO-WAN** perturbs point and box prompts with zero-mean Gaussian noise. Other real-world biases—e.g. inconsistent text queries across annotators—are not modeled, and could alter the observed failure modes of SAM or Grounded-SAM.

This work *measures* robustness; it does not propose a noise-aware training algorithm. Consequently, conclusions about "limitations of current LNL techniques" are empirical, not prescriptive.

Finally, because the benchmark re-uses publicly available images, we do not study privacy leakage or disparate performance across demographic groups.

# 6 Discussion

Our experiments demonstrate that label noise—whether from imprecise human annotations, automated tools, or weak prompts—can substantially degrade the performance of instance segmentation models. We introduced both synthetic and weakly annotated benchmarks that systematically capture real-world noise patterns, ranging from boundary misalignments to class confusion and missing instances. Even moderate levels of noise can erode confidence in model predictions and lead to notable mAP reductions, highlighting the sensitivity of current architectures to spatial inaccuracies.

In particular, our results show that (1) models trained on large datasets like COCO and Cityscapes are far from robust under moderate noise, exhibiting over 10% drops in mask mAP, (2) scale noise severely mislead boundary-based metrics, and (3) while prompt-based foundation models reduce labeling effort, they also introduce new biases, and themselves are not fully immune to noisy prompts. These outcomes underscore the gap between current label-noise handling strategies—mostly devised for image classification—and the complexities of segmentation tasks, where spatial quality is paramount.

## 6.1 Confidence and Loss Analysis

Our study reveals that various architectures and backbones exhibit sensitivity to noise, impacting not only mask quality but also confidence in instance identification. As illustrated in Figure 7b, increased label noise correlates with diminished confidence in model predictions, underscoring the vulnerability of different model architectures to labeling accuracy.

This reduction in confidence is further evidenced in Figure 10, where increased label noise results in poorer mask quality and reduced confidence in the classification head.

We examine the model's ability to distinguish noisy from clean annotations. Figure 8 shows two experiments: in the first, 40% of instances contain class noise; in the second, 40% have medium-level spatial noise. Under class noise, the model's classification losses form two roughly distinct Gaussian distributions, suggesting partial separation of clean and noisy samples. By contrast, when spatial noise is introduced, the losses remain intermixed throughout training. This highlights the challenge of boundary-level label errors for methods relying on loss-based separation. Further experimental details and additional results on learning with noisy labels appear in the supplementary.

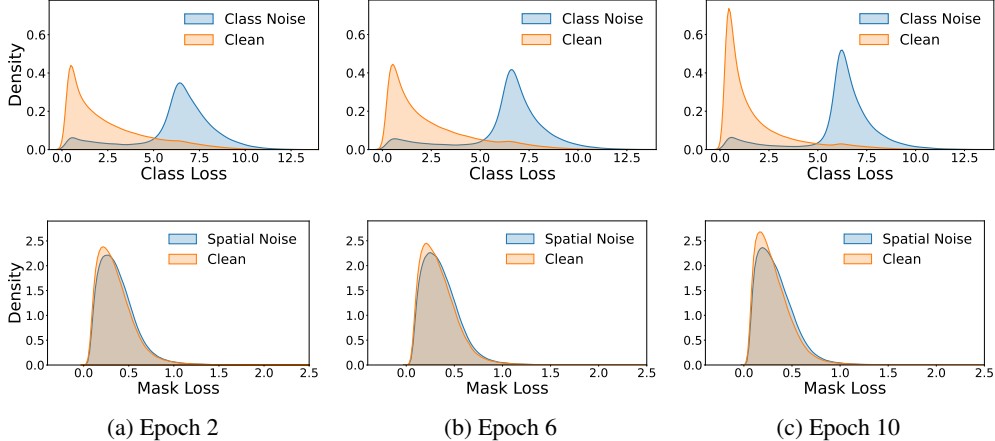

Figure 8: Class and Mask Loss Distribution of Mask-RCNN (R50) trained on COCO easy benchmark at different epochs during training.

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

## A  Ejection Fraction Analysis in the CAMUS Dataset

The CAMUS dataset Leclerc et al. [2019] provides 2D echocardiographic images along with high-quality, expert-annotated labels of the left ventricle (LV). A critical clinical metric in these annotations is the left ventricle's *ejection fraction* (EF), defined as:

$$\text{EF} = \frac{\text{EDV} - \text{ESV}}{\text{EDV}} \times 100\%, \tag{1}$$

where EDV is the end-diastolic volume (i.e., the LV volume at its most dilated state) and ESV is the end-systolic volume (the LV volume at maximal contraction). EF offers a succinct quantification of cardiac pump efficiency: a healthy range is typically considered to be above 50%, while borderline or reduced EF can indicate impaired cardiac function.

**Clinical Implications and Risks.**  Misestimations of the LV boundary—especially at the end-diastolic or end-systolic frames—can propagate into disproportionate errors in volume computations. Even small annotation noise around the boundary may shift the EF from borderline-normal (e.g., 45%) to a clearly abnormal ($\approx 39\%$) or misleadingly normal ($\approx 50\%$) reading. Such inaccuracies pose a risk for misdiagnosis or delayed therapeutic intervention, since EF underlies critical clinical decisions, including the prescription of certain medications, lifestyle interventions, or further diagnostic procedures.

**Noise-Induced Errors.**  Figure 9 (to be added) illustrates how a noisy annotation around the LV boundary at end-diastole can lead to an overestimation or underestimation of EDV. When combined with an equally skewed ESV, the net EF deviation can be clinically significant. We examine morphological dilation of the ESV boundary, along with moderate localization noise in both EDV and ESV, using the "low" noise setup described in the main text.

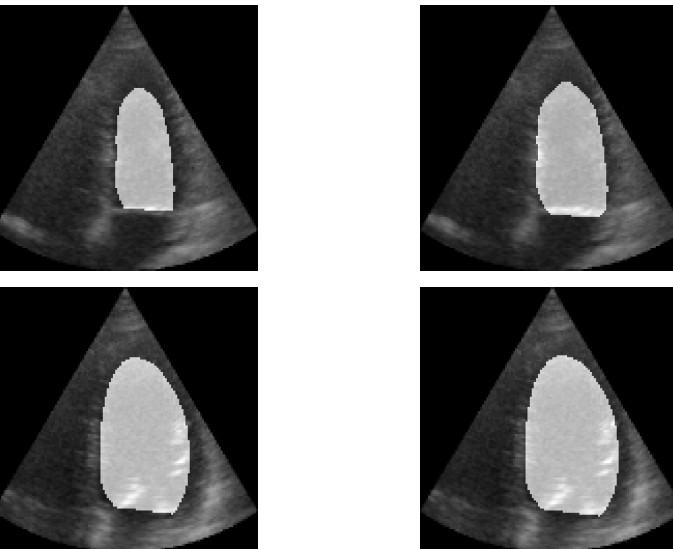

Figure 9: Example of ESV (top) and EDV (bottom) from the CAMUS dataset (left) and their noisy counterparts (right). Even modest boundary distortions can shift EF calculations significantly.

**Evaluation Under Noisy Labels.**  We trained a simple convolution-based U-Net model, as described in Leclerc et al. [2019], on both **clean** and **noisy** CAMUS annotations, and compared the results in Table 6. Evaluation metrics are **Dice Index** for segmentation overlap of the left ventricle (LV) at end-systolic (ES) and end-diastolic (ED) frames, **EF Error** as mean absolute error compared to 2D compute of EF values from the labels in percentage points (p.p), as well as **HD** (Hausdorff Distance) for boundary alignment.

As Table 6 indicates, the model trained on noisy labels tends to yield worse Dice overlap and a higher EF error than when trained on clean labels, underscoring the sensitivity of medical diagnostics to

Table 6: Comparing UNET results on clean vs noisy CAMUS data.

| Training Data | Dice (%) ES | Dice (%) ED | EF Error (p.p.) | HD (mm) (ED frame) |
|---|---|---|---|---|
| Clean | 86.9 | 91.1 | 2.1 | 6.3 |
| Noisy | 82.1 | 87.5 | 4.5 | 11.25 |

Table 7: Ablate the performance evaluation of Mask R-CNN with Spatial label noise across all data on **COCO-N**.

| Severity Metric | Low mAP | Low B-mAP | Medium mAP | Medium B-mAP | High mAP | High B-mAP |
|---|---|---|---|---|---|---|
| Clean | 34.6 | 20.6 | 34.6 | 20.6 | 34.6 | 20.6 |
| Dilation | 32.8 | 18.5 | 29.1 | 14.2 | 26.4 | 10.3 |
| Erosion | 29 | 15.7 | 22 | 9.5 | 17.4 | 5.3 |
| Opening | 34.6 | 20.7 | 34.7 | 20.7 | 34.6 | 20.6 |
| Random Scale | 34.1 | 20.4 | 32.4 | 18.5 | 30.8 | 17.1 |
| Shifting | 28.2 | 15.4 | 26.6 | 14.0 | 21.1 | 8.6 |
| Localization | 34.4 | 20.4 | 34.2 | 20.1 | 33.5 | 19.4 |
| Approximation | 34.7 | 20.8 | 32.5 | 18.8 | 30 | 16.3 |

annotation precision. Crucially, this discrepancy demonstrates that even modest boundary errors can propagate into clinically important EF ranges, highlighting the urgency of robust noise-handling strategies in echocardiographic segmentation tasks.

# B   Additional Experiments

Figure 11 compare the mAP and boundary mAP of original vs. noisy annotations. The top row illustrates the morphological operations used for scale-based spatial distortion, while the bottom row shows the specific noise types we apply in our benchmark.

Table 8: Evaluation results of instance segmentation models (Boundary mAPCheng et al. [2021a]) under various noise levels.

| Dataset | Model | Clean | Easy | Medium | Hard |
|---|---|---|---|---|---|
| COCO-N | M-RCNN (R50) | 20.6 | 18.9 | 17.5 | 16.3 |
| | M-RCNN (R101) | 22.2 | 20.4 | 19.0 | 17.4 |
| | M2F (R50) | 30.0 | 28.6 | 26.7 | 23.8 |
| | M2F (Swin-S) | 32.6 | 30.9 | 29.3 | 26.2 |
| | YOLACT (R50) | 15.7 | 14.4 | 13.5 | 12.4 |
| Cityscapes-N | M-RCNN (R50) | 33.4 | 28.4 | 24.7 | 22.8 |
| | M-RCNN (R101) | 34.3 | 30.7 | 29.0 | 25.4 |
| | YOLACT (R50) | 16.5 | 16.5 | 14.5 | 13.3 |

To further validate our noise design choices and their impact, we obtained additional experiments. As presented in Table 9, we evaluated the traditional symmetric and asymmetric class noise on instance segmentation using MASK-RCNN with two different backbones to assess the resulting performance

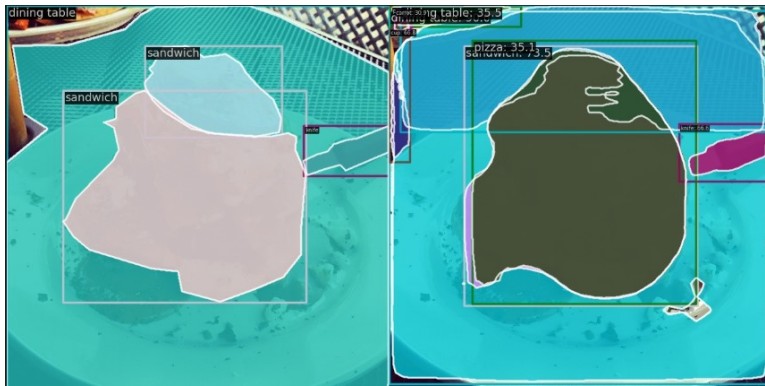

Figure 10: Visual results of Mask-RCNN using the **COCO-N** easy benchmark. Since the model is uncertain it observe different objects (pizza and sandwich in the bottom image) fooling the NMS operation.

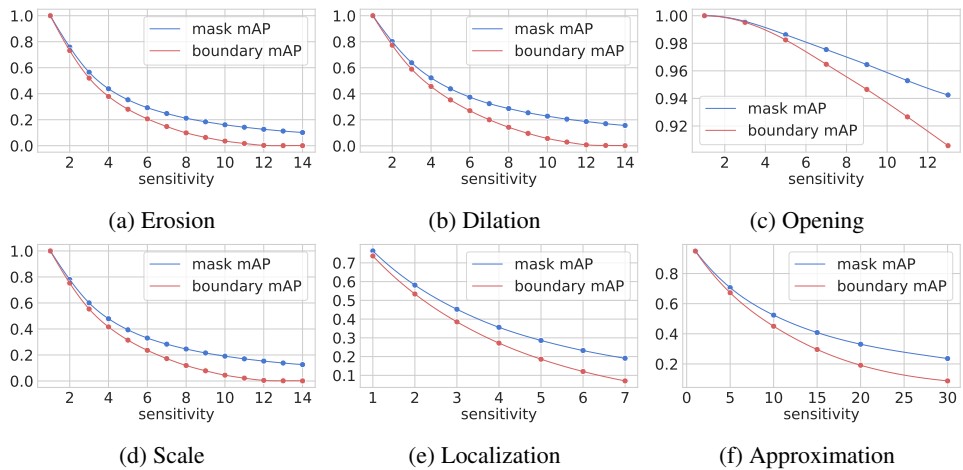

Figure 11: The mAP and boundary-mAP metrics between real annotations from COCO dataset and their **COCO-N** annotations counterpart.

400  degradation. "Sym $p\%$" refers to symmetric class confusion with probability $p$, while "Asym $p\%$"
401  denotes mislabeling concentrated in a smaller set of classes Natarajan et al. [2013], Xiao et al. [2015].

Table 9: Class noise ablation reporting $mAP^{\text{box}}$ and $mAP^{\text{mask}}$

| Models/Labels | Clean | Sym 20% | Sym 50% | Sym 80% | Asym 40% |
|---|---|---|---|---|---|
| M-RCNN (R50) | 38/34.6 | 35.5/31.9 | 32.2/29.2 | 22.5/20.2 | 34.6/31.4 |
| M-RCNN (R101) | 40.1/36.2 | 37.5/33.6 | 34.5/31 | 25.2/22.7 | 36.8/33.2 |

402  Next, we examined the affects of label noise and the additional impact of spatial noise on mask quality,
403  as shown in Table 10. We assessed the quality of all masks through the foreground-background
404  segmentation task of a trained model. The results indicate that the mask quality deteriorates more
405  significantly when spatial noise is incorporated along with traditional class noise.

406  In addition to evaluating the benchmark itself, we extended our analysis to include the impact on
407  object detection performance. Specifically, we examined the Boundary $- mAP$ and $mAP^{\text{box}}$ scores,
408  as presented in Tables 8 and 11 respectively. This tables highlights the detrimental effects of spatial
409  label noise on the boundaries of the masks, as well as bounding box quality, in addition to the
410  previously discussed impacts on mask quality. By analyzing the $mAP^{\text{box}}$, we aim to demonstrate
411  the broader implications of our noise design choices, showing that spatial noise not only affects
412  segmentation masks but also significantly degrades the performance of object detection tasks. This

Table 10: Foreground-background segmentation results under class and spatial noise. The symbol "+" indicates an added spatial corruption using M-RCNN(R50).

| Foregound noise | bbox | segm | boundry |
|---|---|---|---|
| clean | 42 | 35.8 | 22.4 |
| 20 % | 40.7 | 34.9 | 21.7 |
| 20 % + Easy | 40.4 | 34.2 | 21.2 |
| 30% + Medium | 39.6 | 32.7 | 19.9 |
| 40% + Hard | 38.7 | 30.6 | 18.3 |
| 50 % | 38.7 | 32.7 | 20.7 |

comprehensive evaluation underscores the robustness of our benchmark in assessing the performance degradation across different aspects of instance segmentation and object detection.

Table 11: Evaluation Results of Instance Segmentation Models under Different Benchmarks reporting $AP^{box}$.

| Dataset | Model | Clean | Easy | Medium | Hard |
|---|---|---|---|---|---|
| COCO-N | M-RCNN (R50) | 38 | 35.4 | 34.3 | 33.4 |
| | M-RCNN (R101) | 40.1 | 37.4 | 36.5 | 35.2 |
| | M2F (R50) | 45.7 | 42.2 | 43.7 | 44.7 |
| | M2F (Swin-S) | 49.3 | 47.9 | 47.1 | 45.7 |
| | YOLACT (R50) | 30.8 | 29.2 | 28.2 | 27.7 |
| Cityscapes-N | M-RCNN (R50) | 41.5 | 35.7 | 32.8 | 31.2 |
| | M-RCNN (R101) | 39.8 | 32.8 | 29.6 | 26.8 |
| **COCO-WAN** | M-RCNN (R50) | 36.3 | 34.1 | 25.5 | 22.4 |

Finally, we present results on the long-tailed segmentation dataset LVIS, as shown in Table 12. The findings reveal a significant impact, with a 50% reduction in boundary IoU under the hard benchmark conditions. This provides evidence of an exacerbated effect in long-tailed scenarios, highlighting the increased challenges posed by our noise design in datasets with imbalanced class distributions.

Table 12: Performance on LVIS-N (Mask R-CNN R50-FPN). We report mAP / Boundary mAP under various noise levels.

| Dataset | Clean | | Easy | | Medium | | Hard | |
|---|---|---|---|---|---|---|---|---|
| | $AP$ | $AP^B$ | $AP$ | $AP^B$ | $AP$ | $AP^B$ | $AP$ | $AP^B$ |
| LVIS-N | 22.8 | 22.1 | 15.5 | 14.3 | 17.7 | 13 | 13.3 | 11.2 |

## C  Additional Noise Visualizations

Figure 12 presents additional samples from our benchmark under different intensities of spatial label noise. Each row highlights a specific set of distortions—such as boundary approximations or morphological operations—applied to one or more instances. As the noise severity increases from left to right, the object contours become visibly degraded, illustrating the range of realistic annotation errors our benchmark can simulate.

Figure 15 provides a qualitative comparison of Mask R-CNN and Mask2Former under varying noise levels. The top row shows both models' predictions on a clean COCO image: each accurately delineates the car and surrounding objects with sharp, well-aligned masks. In the middle row, Mask R-CNN is challenged by the Easy, Medium, and Hard variants of COCO-N—its masks become

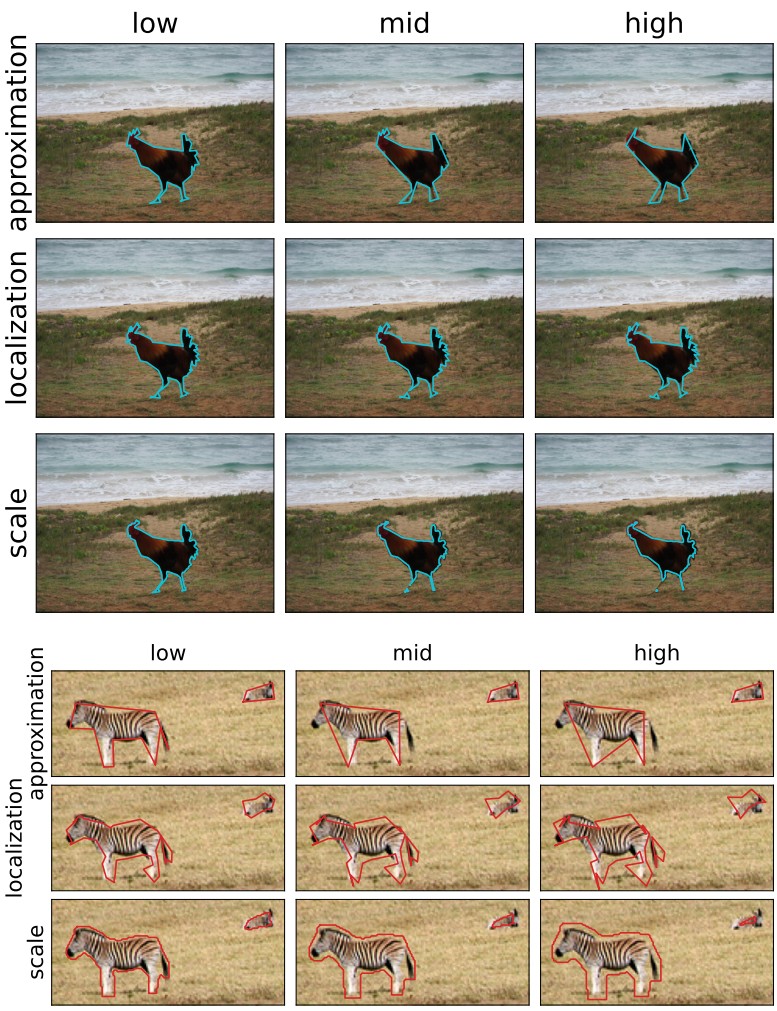

Figure 12: Additional illustrating the effects of the spatial noises on one or two instances with various scales, similar to fig. 2

progressively fragmented, with missing segments and increasingly jagged boundaries. The bottom row reveals that Mask2Former, while initially more robust, also suffers under stronger noise: its Easy predictions remain close to the clean baseline, but Medium and Hard noise lead to boundary bleed and partial omissions. Overall, this figure illustrates that spatial annotation errors systematically degrade mask quality in both CNN- and transformer-based architectures, with severity correlating with noise intensity.

# D   Implementation Details

This section elaborates on the architectures, datasets, noise definitions, and the levels of asymmetric noise used in our experiments. We also detail the noise intensity applied in the benchmark, along with the hardware configurations and convergence times.

## D.1   Architectures

We explore the effects of label noise on various instance segmentation models, encompassing multi-stage (Mask R-CNN He et al. [2017]), single-stage (YOLACT Bolya et al. [2019]), and query-based (Mask2Former Cheng et al. [2021b]) architectures. To achieve a comprehensive analysis, we experimented with different feature extractors, we used convolutional backbones such as ResNet-

444 50 He et al. [2015] for all models and ResNet-101 for Mask R-CNN, alongside a transformer-
445 based backbone (Swin-B Liu et al. [2021]) for Mask2Former. For the integration of multi-scale
446 features, Feature Pyramid Networks (FPN) Lin et al. [2016] were employed across all models
447 except Mask2Former, which utilizes Multi-Scale Deformable Attention (MSDeformAttn) Zhu et al.
448 [2020], as multi-scale feature representation. All models and configurations implementations from
449 MMDetection Chen et al. [2019c].

## D.2 Datasets

451 **COCO**   dataset for training and evaluating algorithms that segment individual objects within a
452 scene. It contains about 330,000 images, annotated with over 1.5 million instances masked from 80
453 categories that are also part of 12 super-categories.

454 **Cityscapes**   dataset is designed for training and evaluating algorithms in urban scene understanding,
455 particularly for segmentation tasks. It comprises a collection of images captured in 50 different cities,
456 featuring 5,000 annotated images with 19 classes for evaluation, covering a range of urban object
457 categories such as vehicles, pedestrians, and buildings.

458 **VIPER**   VIPER Richter et al. [2017] is a synthetic dataset generated from the GTA V game engine.
459 It provides per-pixel annotations for a broad range of 31 categories in photorealistic urban scenes,
460 making it ideal for benchmarking under controlled conditions. Because VIPER annotations are
461 automatically rendered (rather than hand-labeled), they are virtually free from human annotation
462 errors, allowing precise evaluation of how injected label noise affects segmentation performance.

463 **LVIS**   dataset is based on COCO images and curated to provide a comprehensive benchmark for
464 instance segmentation, emphasizing rare object categories. It contains over 2 million high-quality
465 instance annotations across 1,203 categories, making it one of the largest and most diverse datasets
466 for instance segmentation. The LVIS dataset is particularly noted for its long-tail distribution of
467 object categories, which poses significant challenges for segmentation algorithms and help us to asses
468 the abilities of segmentation algorithms to deal with label noise in this scenario.

469 .

## D.3 Hardware details

471 MS-COCO based experiments (include both COCO and LVIS) and VIPER conducted on local
472 machine with 4 Nvidia RTX A6000 or 4 Nvidia RTX 3090, ranging from 20 hours (Mask-RCNN
473 with R50) to 7 days (Mask2Former with SWIN transformer beckbone), training for 12 epochs for
474 all models except YOLACT that trained for 50 epochs. Cityscapes experiments conducted on local
475 machine with one instance of Nvidia RTX 3090, training for 12 epocs for about 12 hours. All
476 experiments use the default configs from MMDetection Chen et al. [2019c].

# E   Learning with Noisy Labels

478 As described in the paper, class noise is separable, allowing one to derive noisy instances from clean
479 ones (refer to Figure 13a). However, dealing with mask losses is more challenging. The loss of noisy
480 instances consists predominantly of correctly labeled pixels, with only a few noisy ones (refer to
481 Figure 13b). Furthermore, since most spatial noise occurs at the boundaries, these areas are where
482 the model exhibits the least confidence Kimhi et al. [2024]. This complexity makes it impossible to
483 distinguish between pixel-level noisy and clean data, posing a significant challenge in developing a
484 spatial noise solution to learn from noisy labels.

485 Due to these difficulties, we compared a class noise method to handle noisy labels. Table 13 presents
486 the results on the COCO-N benchmark, comparing standard Cross-Entropy with Symmetric Cross-
487 Entropy Wang et al. [2019]. While there is a marginal improvement, the method still faces challenges
488 as the noise level increases.

Table 13: Evaluation Results of Instance Segmentation with different losses learning with noisy labels trained on COCO-N dataset (mAP / Boundary mAP).

| **Loss** | Clean | | Easy | | Mid | | Hard | |
|---|---|---|---|---|---|---|---|---|
| | $AP$ | $AP^B$ | $AP$ | $AP^B$ | $AP$ | $AP^B$ | $AP$ | $AP^B$ |
| CE | 34.6 | 20.6 | 31.8 | 18.9 | 30.3 | 17.5 | 28.4 | 16.3 |
| SCE | 32.5 | 19.5 | 32.1 | 18.9 | 30.8 | 17.8 | 28.8 | 16.4 |

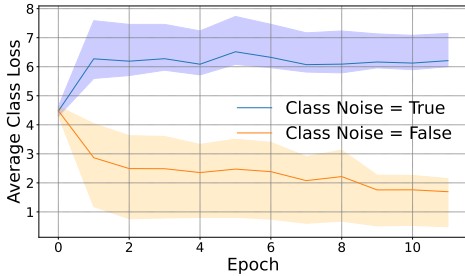 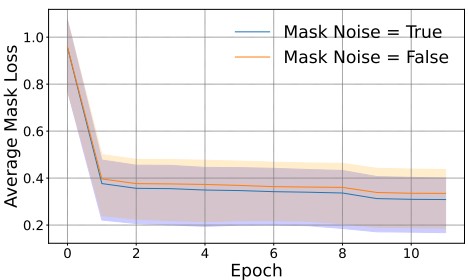

(a) Class Loss Separation. Average of the class loss of the Coco dataset, with the 25% and 75% quantiles as margins - per epoch of training.

(b) Mask Loss Separation. Average of the mask loss of the Coco dataset, with the 25% and 75% quantiles as margins - per epoch of training.

## F SAM Finetune with label noise

Since for weakly supervision annotations we heavily relay on SAM Kirillov et al. [2023], we exemine how noise in prompt effect the model itself in two setups, *zero-shot*, that corespond to the quality of the masks produced by sam, and *fine-tuning*, as a popular paradigm of using SAM for a downstream application. For the zero-shot, we prompt SAM with the grounded bounding boxes of the validation set of COCO as well as noisy boxes with the **COCO-N hard** type of noise on the validation annotations. For fine-tuned, we exemine fine tuning with both clean and noisy **COCO-N hard** annotations masks. Table 14, shows both mIoU and F1 scores of the masks produced by SAM, showing that the quality of masks can be increased when fine-tuned, compared with zero shot training with high quality prompts. Fine-tuning with noisy annotations however, is less sever, when prompting with cerfully designed prompts, compared to noisy prompts. Our findings suggest that the quality of prompts are fur more important then the qua

Table 14: Evaluation of prompt Instance segmentation on SAM

| **Annotations Method** | Clean | | COCO-N Hard | |
|---|---|---|---|---|
| | $IoU$ | $F1$ | $IoU$ | $F1$ |
| Zero-shot | 79.78 | 87.49 | 67.99 | 63.30 |
| Fine-tune | 79.91 | 78.6 | 77.47 | 76.18 |

## G Biases of Self-annotating Datasets

More visual results of the weakly supervised annotations created by SAM are presented in Figure 14. A significant number of annotations were curated by this process (top row), reducing label noise, particularly in cases where the original annotations suffered from approximation noise. In other instances, where an object is surrounded by similar colors or illumination conditions, the annotations become noisier around the boundaries, exhibiting weak localization noise (middle row).

The specific context of the dataset annotations can influence what the user is looking for. We observed cases where there is ambiguity in the definition of certain objects, such as stove-tops (bottom row).

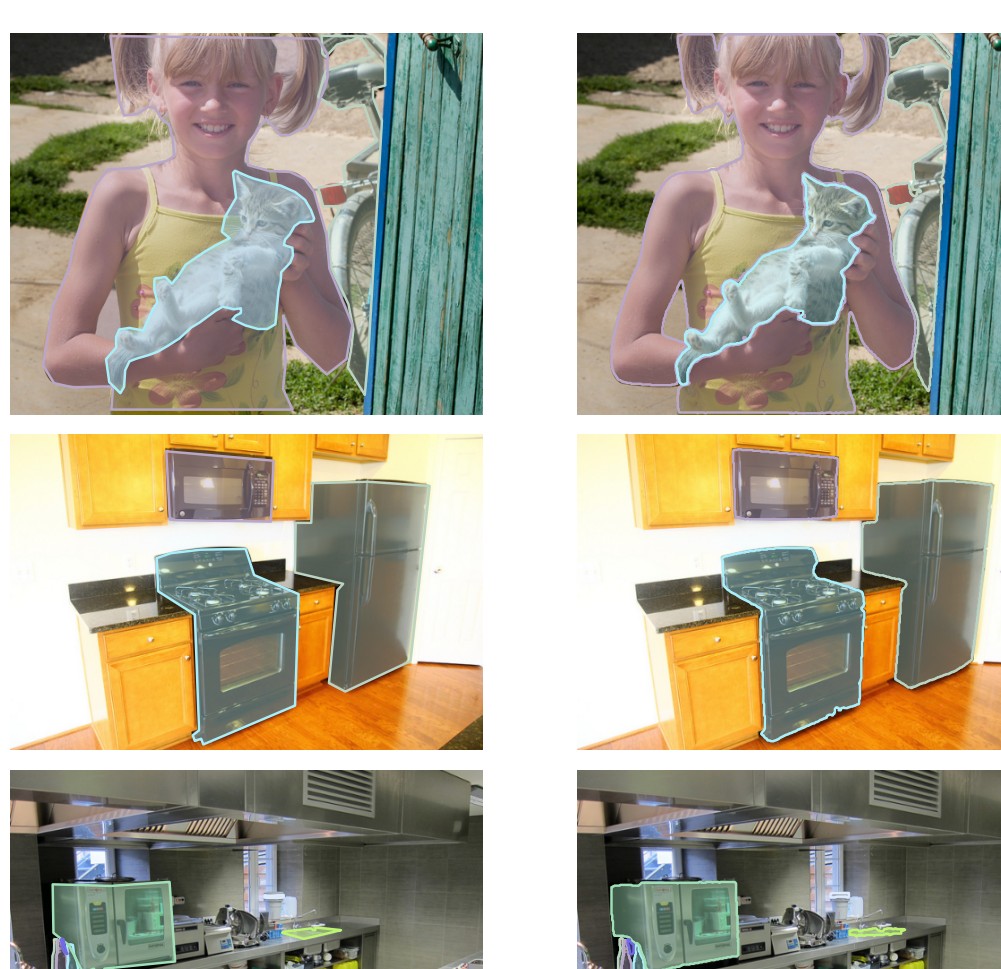

Figure 14: Pairs of COCO annotations (left) and **COCO-WAN** easy annotations (right). Top pair shows high fidelity annotations for **COCO-WAN**, compared to the original noisy counterparts. The bottom example examine that when color changes by little, even with bounding box prompts, SAM confuses due to color biases in segments and can not capture the desired segments such as stovetop or sink.

While SAM is familiar with the concept of a stove-top, it lacks the contextual knowledge of what it should be within the specific context of the COCO dataset, leading to poor masking.

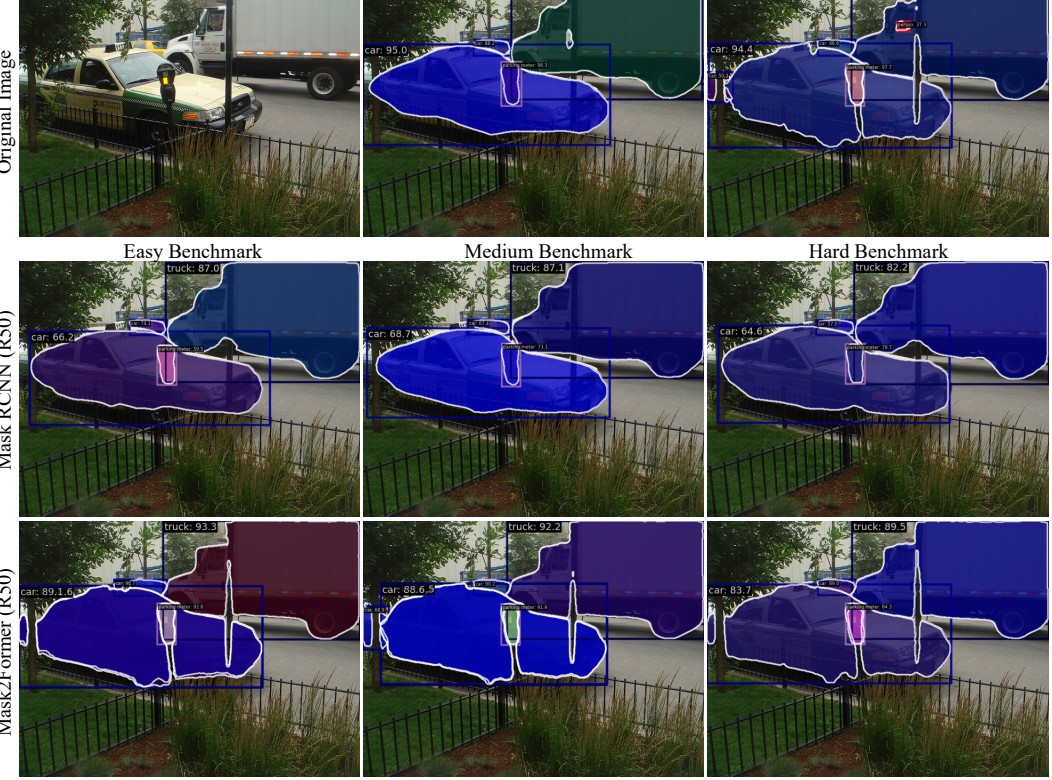

Figure 15: Comparison Mask RCNN and Mask2Foramer models predictions. Top row (left to right): original image M-RCNN and M2F predictions on clean COCO. Middle: M-RCNN predictions on Easy, Medium and Hard **COCO-N**. Bottom: M2F predictions on Easy, Medium and Hard **COCO-N**.

