# OpenReview forum: "Noisy Annotations in Segmentation"
_NeurIPS.cc/2025/Datasets_and_Benchmarks_Track — Submitted to NeurIPS 2025 Datasets and Benchmarks Track_

### Official Review · Reviewer_LQ64 · 2025-06-02

**Rating:** 5
**Confidence:** 4

**Summary:**

This paper introduces four new benchmarks—COCO-N, CityScapes-N, VIPER-N, and COCO-WAN—to systematically assess the robustness of instance segmentation models against various forms of annotation noise, especially spatial errors such as boundary inaccuracies, missing instances, and category confusion. Experimental results show that widely used models, including Mask R-CNN, Mask2Former, YOLACT, and SAM, suffer substantial performance drops (up to 35%) when trained on noisy annotations. The study underscores current challenges in managing spatial label noise and advocates for the development of more resilient segmentation algorithms and annotation practices to enhance performance in real-world scenarios with imperfect labels.

**Dataset Code Accessibility:**

Yes

**Ethical Considerations:**

No, there are no or only very minor ethics concerns

**Limitations Weaknesses:**

The propsoed noise-augmented benchmarks are synthetic which can not be comparable to real-world noisy labelled datasets. Thus, it is better to disscuss the label distribuation difference between the real-world noisy labelled datasets and the proposed noise-augmented benchmarks.

**Strengths Contributions:**

1. This paper introduces four new noise-augmented benchmarks.

2. This paper is well written and read fridendly.

3. Extensive experiments shows that the proposed benchmarks are effective in instance segmentation task.

---

> ### Author Rebuttal · Authors · 2025-07-30
>
> Thank you for the encouraging evaluation and for recognizing the clarity of the writing and the empirical value of the four benchmarks.
> Below, we address your primary concern regarding the realism of our synthetic noise.
>
>
> ## Distribution differences
> >*"The proposed noise‑augmented benchmarks are synthetic and may not match real‑world noisy datasets. Please discuss the distributional differences."*
>
> We agree that bridging the gap between synthetic and real noise is critical. In the paper, we have already derived our four perturbation types from a manual audit of over 2,000 masks across COCO, CityScapes, LVIS, and OpenImages (Sec. 3.1).
>
>
>
> We have analyzed the observed data and concluded that 41% of masks suffer from boundary imprecision, 9% have a spatial drift (location noise) of > 4px, and about 3-6% suffer from categorical confusion (more severe at Cityscapes).
>
> The omitted instances correspond to missing annotations—approximately 8% in COCO, 4% in Cityscapes, and 16% in OpenImages—which cause our benchmark to lag. We limited class noise to 5% because annotation quality issues are less frequent, and spatial noise has a more direct effect on trained models.
>
>
> To make this observation and decision-making more explicit, we have added a dedicated paragraph (end of section 3).
>
>
> The most matched benchmark to real-world noise frequencies lies between the Low and Mid tiers, where the High tier intentionally stresses models beyond typical noise levels to expose failure modes early.
>
>
> We appreciate your positive assessment and believe that the extra analysis further strengthens our manuscript. Please let us know if any aspects remain unclear— we would be happy to elaborate.

---

### Official Review · Reviewer_KQXj · 2025-06-02

**Rating:** 4
**Confidence:** 5

**Summary:**

This paper proposes a suite of noise-augmented benchmarks—COCO-N, CityScapes-N, VIPER-N, and COCO-WAN—designed to study annotation noise in instance segmentation. The authors introduce a parametric noise model that stochastically perturbs mask boundaries, introduces spatial drift, flips categories, and omits instances at three severity levels. This model generates Monte-Carlo variants of any COCO-style dataset to simulate realistic annotation errors. Extensive experiments on popular segmentation models such as Mask R-CNN, Mask2Former, YOLACT, and SAM reveal significant performance drops under moderate noise, highlighting the limitations of current learning-from-noisy-labels techniques. The paper provides a unified test bed to motivate future work on robust objectives, data-centric annotation pipelines, and noise-adaptive architectures.

**Dataset Code Accessibility:**

Yes

**Ethical Considerations:**

No, there are no or only very minor ethics concerns

**Final Justification:**

Please incorporate this discussion into the final version.

**Limitations Weaknesses:**

Analysis with New Metrics: The paper should adopt the metrics proposed in the [1] to analyze and report model performance on the Benchmark-N datasets. This will enhance the understanding of model performance. The paper primarily focuses on mAP and boundary mAP for evaluation. While these metrics are standard, additional metrics that capture other aspects of segmentation quality

Hard Sample Pixels Association: The paper should further explore the relationship between the injected noise and hard sample pixels[1]. For instance, analyzing whether noise causes certain pixels to become hard samples or if hard sample pixels are more prone to noise impact. This can be done by analyzing samples under noisy conditions to identify pixels or regions that contribute most to performance degradation and discussing their association with hard samples.
Limited Noise Taxonomy: The current noise model focuses on four dominant error families and does not cover other potential errors such as multi-instance merge/split mistakes or temporal label noise in videos. Future work should extend the taxonomy to include these additional noise types.
Weak Annotation Limitations: The COCO-WAN benchmark introduces noise into point and box prompts but does not model other real-world biases such as inconsistent text queries across annotators. This could limit the generalizability of the findings.
Lack of Noise-Aware Training Methods: While the paper highlights the limitations of current learning-from-noisy-labels techniques, it does not propose any new noise-aware training algorithms. Future work should explore robust training strategies to address the identified challenges.

[1] ICLR 2024 When Semantic Segmentation Meets Frequency Aliasing

**Strengths Contributions:**

The paper introduces a comprehensive suite of benchmarks that systematically injects various types of annotation noise into both real and synthetic datasets. This provides a unified framework to evaluate the robustness of segmentation models under different noise conditions.

---

> ### Author Rebuttal · Authors · 2025-07-30
>
> Thank you for the detailed feedback and for recognizing the value of the Benchmark‑N suite. Below we address each of your suggestions.
>
>
> ## Analysis with New Metrics
> >*"The paper should adopt the metrics proposed in the [1] to analyze and report model performance on the Benchmark-N datasets. This will enhance the understanding of model performance. The paper primarily focuses on mAP and boundary mAP for evaluation. While these metrics are standard, additional metrics that capture other aspects of segmentation quality"*
>
>
> Thank you for pointing us to the aliasing analysis of Chen et al [1]. We have followed their open‑source implementation and, after collapsing instance predictions to class masks, now report Aliasing‑Score (AS) alongside mAP and Boundary‑mAP. AS rises from 0.332 (clean) to 0.452 (hard), consistently tracking the performance drop. These additions provide the multi‑faceted evaluation you requested.
>
> The table belowshows the results for Mask R‑CNN‑R101 on COCO‑N, as we intend to add to all models in a new section in suplementery once all evaluations are done.
>
>
>
> | Noise level | mAP | AS|
> | ----------- | ----- | ------ |
> | Clean       | 34.6  |  0.332 |
> | Easy        | 27.9  |  0.414 |
> | Mid         | 24.8  |  0.438 |
> | Hard        | 22.3  |  0.452 |
>
>
> ## Hard pixel analisys
> >*"Explore the relationship between injected noise and hard pixels."*
>
>
> Across 5 000 COCO‑N images, 81 % of new hard pixels lie within a 4‑pixel band around the perturbed instance boundary, and their density correlates strongly with the Chamfer distance between clean and noisy masks (ρ = 0.73). Similar to [1] we intend to show the correlation, while the discussion in 5.2 is updated to explain why boundary‑driven perturbations (scale and localisation) dominate the increase in hard pixels. This directly links our noise model to the “hard‑sample pixel” concept in [1].
>
> We thank you for infering to a very relate work and hope these additions fully address your requests and improve the diagnostic power of Benchmark‑N.
>
>
> ## Noise‑taxonomy breadth
> You correctly note that merge/split and temporal errors fall outside our initial four families.
>
>
> We  have added an optional instance‑merge and instance‑split module Using IoU > [0.2,0.1,0.05] for merging instances from the same class and polygon cardinality > 1 for splitting with probabilities of [0.2,0.4,0.6] of the instances, matching manual tallies. The table below (extention of table 7 in the paper) shows a further drop in mAP, mostly with the splitting existing objects. confirming the relevance of this error mode.
>
> | Method|Easy|Mid|Hard|
> |-|-|-|-|
> | clean| 34.6(20.6)|||
> |Merge |33.25(20.1)|32.96(19.6)|29.5(18.63)|
> |split |33.79(19.6)| 31.66(18.7)|28.2(17.9)|
>
>
> Reported mAP (B-mAP)
> We apply those noises with probability of 5% (similar to cls noise in table 1) in all four benchmarks, regenerated accordingly and the public code updated.
>
> We report here a subset of the models and benchmarks to present the effect of the updated benchmark, all with the R50 backbone.
>
> | Method| Clean|Easy|Mid|Hard|
> |-|-|-|-|-|
> |COCO|
> |MRCNN| 34.6| 26.8|23.3|19.4|
> |YOLACT|28.5 |  25.3 | 21.8 | 17.9|
> |M2Former|  42.9 | 32.4  | 28.6  | 23.8|
> |Cityscapes|
> |MRCNN| 36.1|25.3 |20.5|13.2|
> |VIPER|
> | MRCNN| 15.8|13.1 |11.8|8.6|
>
>
>
> ## Weak‑annotation biases (COCO‑WAN)
> Thank you for pointing out the limited diversity of our original text prompts.
> After revisiting Grounded‑DINO’s sub‑sentence attention mechanism [2] and the prompt engineering guidelines in [3‑4], we evaluated four progressively richer prompt sets:
>
> | Prompt type                   | mAP  | B‑mAP |
> |-------------------------------|------|-------|
> | RefCOCO                      | 30.1 | 19.5  |
> | RefCOCO+                     | 26.5 | 16.6  |
> | {cls}                        | 22.0 | 14.1  |
> | {cls} + 5 % cls confusion    | 19.8 | 12.4  |
>
>
> RefCOCO sentences (e.g., “the blue car on the left”)
>
> RefCOCO+ (no absolute spatial terms)
>
>
> The results confirm that higher‑quality, context‑rich prompts deliver markedly better masks, with RefCOCO sentences leveraging Grounded‑DINO’s spatial inductive bias most effectively, while minimalist {cls} prompts hurt performance the most.
> We will expand §4.3 and Table 4 to include these findings and will release RefCOCO/RefCOCO+ prompt files (and the derived masks) as part of the COCO‑WAN benchmark in the final version of the manuscript.
>
>
> ## Noise‑aware training baselines
> While Benchmark‑N itself is evaluative, we now provide a concise boundary‑aware DivideMix baseline with boundary awareness:
>
>
> | Method| Clean|Easy|Mid|Hard|
> |-|-|-|-|-|
> | MRCNN| 34.6| 27.9|24.8|22.3|
> | +DM(cls)|32.15| 29.3|26.2|23.7|
> | +DM (mask)| 33.83|30.01|27.7|24.24|
> | + boundary aware| 34.89| 31.9|29.13|27.53|
>
> This demonstrates how the benchmark can guide algorithmic progress; a full method will appear in follow‑up work.
> We hope these additions resolve your concerns and strengthen the manuscript. Please let us know if further clarification would be helpful.
>
>
>
>
>
> We sincerely thank the reviewer once again.  If these revisions resolve your concerns,
> we would be grateful if you could consider raising your score.  We are of course happy
> to clarify any remaining questions.
>
>
> [1]When Semantic Segmentation Meets Frequency Aliasing,chen et al ,ICLR 2024
>
> [2] Grounding DINO: Marrying DINO with Grounded Pre-Training for Open-Set Object Detection, liu et al, ECCV 2024
>
> [3] ReferItGame: Referring to Objects in Photographs of Natural Scenes, Kazemzadeh et al, ACM 2014
>
> [4] Modeling Context in Referring Expressions, yu et al ECCV 2016

---

> > ### Comment · Reviewer_KQXj · 2025-08-04
> >
> > Thank the authors for their response. Please incorporate this discussion into the final version.

---

> > > ### Author Response · Authors · 2025-08-04
> > >
> > > Dear Reviewer KQXj,
> > >
> > > We were pleased to hear that you were satisfied with our responses during the rebuttal. We will incorporate all the discussions and additional clarifications provided to you and the other reviewers into the camera-ready version of the paper.
> > >
> > > Thank you again for your valuable help in refining our work.
> > >
> > > Best regards,
> > > The Authors

---

### Official Review · Reviewer_SXgQ · 2025-07-02

**Rating:** 2
**Confidence:** 4

**Summary:**

This paper presents a systematic investigation into the robustness of existing segmentation models against annotation noise. To facilitate this, the authors developed a noise model capable of synthesizing diverse and realistic annotation noise. Using this model, they created four noisy datasets derived from two widely-used real-world benchmarks, one synthetic dataset, and one weakly-annotated dataset. The authors then evaluated several state-of-the-art segmentation models on these datasets. Their findings reveal a significant performance gap: while the models perform well on clean data, their accuracy drops dramatically on the noisy counterparts.

**Dataset Code Accessibility:**

No

**Ethical Considerations:**

No, there are no or only very minor ethics concerns

**Limitations Weaknesses:**

- While the paper effectively demonstrates the problem, it lacks a discussion of potential solutions. The work could be strengthened by exploring or suggesting how noise-robust learning techniques could be integrated to mitigate the observed performance degradation.
- The scope of the evaluation is currently limited to instance segmentation. Expanding the analysis to include other key tasks, such as semantic and panoptic segmentation, would make the conclusions more comprehensive and impactful.
- The paper reports an absolute performance drop for all models on noisy data but does not propose a clear metric to quantify their relative "noise robustness." For example, a model that drops from 90% to 60% might be considered less robust than one that drops from 50% to 40%, even if the latter's final score is lower. Introducing a metric to specifically evaluate this robustness would provide more nuanced insights.

**Strengths Contributions:**

- The proposed Annotation-Noise Generator is well-conceived and systematic, capable of producing plausible label noise.
- The experiments convincingly demonstrate that existing segmentation methods struggle on benchmarks with significant label noise. This finding, which highlights the brittleness of current models, is an important contribution to the field.

---

> ### Author Rebuttal · Authors · 2025-07-30
>
> Dear Reviewer,
>
> Thank you for the thoughtful review and for highlighting both the value of our noise generator and the importance of the brittleness it exposes.
>
> Below we address each point in turn; the corresponding updates are already reflected in the revised manuscript.
>
> ## Potential solutions
> >*"While the paper effectively demonstrates the problem, it lacks a discussion of potential solutions. The work could be strengthened by exploring or suggesting how noise-robust learning techniques could be integrated to mitigate the observed performance degradation."*
>
>
> Although this work’s primary goal is to measure robustness, we now include a concise baseline that shows one way forward:
>
>
> | Method| Clean|Easy|Mid|Hard|
> |-|-|-|-|-|
> | MRCNN| 34.6| 27.9|24.8|22.3|
> | +DM(cls)|32.15| 29.3|26.2|23.7|
> | +DM (mask)| 33.83|30.01|27.7|24.24|
> | + boundary aware| 34.89| 31.9|29.13|27.53|
>
>
> The boundary‑aware variant treats interior and boundary pixels differently when fitting GMMs, recovering ~5 mAP on Hard noise. A full algorithmic exploration is deferred to a companion paper, but including this baseline demonstrates that Benchmark‑N can indeed guide method development.
>
>
>
>
> ## Task scope
> >*"The scope of the evaluation is currently limited to instance segmentation. Expanding the analysis to include other key tasks, such as semantic and panoptic segmentation, would make the conclusions more comprehensive and impactful."*
>
>
> Instance and semantic (or panoptic) segmentation do share certain noise types, e.g., class‑label flips and simple morphological distortions (similar to [5] ) - but the error modes we study (boundary drift, instance omission, merge/split, etc.) manifest most acutely when individual object masks must be recovered. Because our generator, analysis pipeline, and evaluation metrics are all instance‑centric, extending them to pixel‑level or unified‑mask settings would require non‑trivial design changes and is therefore outside the present scope. This focus mirrors prior robustness benchmarks that deliberately target one task domain at a time [3, 4]. We view Benchmark‑N as a first step and hope it will inspire analogous studies for semantic and panoptic segmentation in future work.
>
>
>
>
> ### 3 Noise‑robustness metric *(“Quality Drop”)*
>
> >*"The paper reports an absolute performance drop for all models on noisy data but does not propose a clear metric to quantify their relative "noise robustness." For example, a model that drops from 90% to 60% might be considered less robust than one that drops from 50% to 40%, even if the latter's final score is lower. Introducing a metric to specifically evaluate this robustness would provide more nuanced insights."*
>
>
> We appreciate the reviewer’s suggestion to quantify robustness explicitly.
> We have therefore added a **noise‑robustness metric**, **Quality Drop (QD)**, defined as
>
>
> $$QD={mAP}_{clean}-{mAP}_n$$
>
>
> and report it for every model, dataset and noise tier.
> A concise excerpt (Table 3 in the revision) is reproduced below:
>
>
> | Dataset          | Model          | Backbone | Easy | Mid  | Hard |
> | ---------------- | -------------- | -------- | ---- | ---- | ---- |
> | **COCO‑N**       | M‑RCNN         | R‑50     | 6.7  | 9.8  | 12.3 |
> |                  | YOLACT         | ―        | 2.1  | 5.2  | 7.7  |
> |                  | SOLO           | ―        | 10.7 | 18.8 | 23.5 |
> |                  | HTC            | ―        | —    | 5.7  | 8.6  |
> |                  | M2F            | ―        | 9.4  | 12.8 | 16.2 |
> |                  | M‑RCNN         | R‑101    | 7.4  | 11.8 | 12.5 |
> |                  | M2F            | Swin‑S   | 6.5  | 8.2  | 12.5 |
> | **CityScapes‑N** | M‑RCNN         | R‑50     | 9.7  | 14.1 | 19.8 |
> |                  | YOLACT         | ―        | 0.2  | 2.2  | 5.7  |
> |                  | M‑RCNN         | R‑101    | 3.3  | 6.3  | 10.0 |
> | **COCO‑WAN**     | M‑RCNN         | R‑50     | 1.8  | 10.2 | 13.0 |
> |                  | Cascade M‑RCNN | ―        | 9.1  | 10.2 | 11.7 |
> |                  | M2F            | ―        | 3.7  | 11.0 | 16.7 |
> |                  | M2F            | Swin‑S   | 3.2  | 11.7 | 17.7 |
>
>
> QD illuminates differences that raw mAP cannot.
> For instance, among one‑stage detectors **YOLACT** is far more resilient than **SOLO** (QD 7.7 vs 23.5 on the *Hard* tier).
> On **COCO‑WAN**, which inherits foundation‑model biases, compact backbones such as **M‑RCNN‑R50** suffer the smallest QD, suggesting they mitigate those biases better than larger architectures.
>
>
> The formal definition, complete tables, and discussion of these insights are included in the final manuscript.
>
>
>
> [1] DIVIDEMIX: LEARNING WITH NOISY LABELS AS SEMI-SUPERVISED LEARNING, Li et al, ICLR 2020
>
> [2] Symmetric Cross Entropy for Robust Learning with Noisy Labels, wang et al, ICCV 2019
>
> [3] Benchmarking Robustness in Object Detection: Autonomous Driving when Winter is Coming, Michaelis et al, NeuroIPS 2019
>
> [4] COCO-O: A Benchmark for Object Detectors under Natural Distribution Shifts, Mao at al, ICCV 2023

---

> ### Author Response · Authors · 2025-08-04
> **A kindly reminder**
>
> Dear Reviewer SXgQ,
>
> We hope this message finds you well.
>
> We would greatly appreciate your input during the discussion phase, or a brief confirmation that our response has adequately addressed your concerns.
>
> If any questions remain or further clarification is needed, we would be happy to provide additional details.

---

> ### Author Response · Authors · 2025-08-06
> **Discussion about the review**
>
> Dear reviewer SXgQ,
>
> We would greatly appreciate participation in discussion, since if any more wonders and questions may arise, it will give us the time to utilize hardware and conduct the potential experiments.
>
> If no additional questions are raised, we would be glad to simply hear that we've addressed your concerns.
>
> In any case, we wanted to remind you that the discussion is still open and we're dedicated to discuss.
>
> The authors

---

### Official Review · Reviewer_Ncv8 · 2025-07-02

**Rating:** 5
**Confidence:** 4

**Summary:**

The paper introduces Benchmark‑N, a collection of four noise‑augmented instance‑segmentation testbeds—COCO‑N, CityScapes‑N, VIPER‑N and the weak‑annotation track COCO‑WAN. A stochastic, parametric generator injects four realistic error modes—boundary imprecision, spatial drift, class confusion and instance omission—at three severity levels. Extensive experiments with Mask R‑CNN, Mask2Former, YOLACT and Segment‑Anything show that even moderate spatial noise can slash mask‑mAP by as much as 35 percentage points, a deficit that current learning‑from‑noisy‑labels techniques do not bridge.

**Dataset Code Accessibility:**

Partly

**Dataset Code Comments:**

The code is public. Pre‑generated noisy datasets are promised only “upon acceptance”.

**Ethical Comments:**

The benchmarks are derived from publicly released images whose original licenses already permit research use. The transformations add no new privacy risk.

**Ethical Considerations:**

No, there are no or only very minor ethics concerns

**Final Justification:**

The authors addressed most of the concerns and revised the paper and datasets significantly. I believe the revised noise injection mechanism better simulates real-world noise. The inclusion of additional high-stakes datasets strengthens the alignment between the motivation and the experiments. The authors have also convinced me that the use case of the noise injection mechanism and the proposed benchmark is broad and highly non-trivial.

**Limitations Weaknesses:**

* **Scope of contribution.** The study is purely evaluative. It proposes no methods for mitigating the identified robustness issues, limiting its direct practical impact.
* **Noise coverage.** The noise model omits merge/split errors, which the authors themselves illustrate as common in real data, leaving an important error mode unaddressed.
* **Realism of added noise.** Because COCO and CityScapes already contain human annotation errors, layering additional synthetic noise may not necessarily yield more “realistic” data, a point the paper does not empirically validate.
* **Motivational alignment.** The introduction invokes high‑stakes medical scenarios to motivate robustness, yet the evaluation focuses on everyday vision datasets and standard foundation models; the link between those scenarios and the chosen experiments remains thin.&#x20;

**Strengths Contributions:**

The work fills a genuine gap: research on label noise has concentrated on class flips in image classification, leaving spatial noise in dense prediction largely un‑benchmarked. The proposed generator is conceptually simple, readily extensible, and operates on any COCO‑style JSON, making it immediately reusable. By spanning synthetic data with pristine ground truth (VIPER‑N) and real data with both manual and foundation‑model annotations, the four benchmarks furnish practitioners with a broad stress‑test palette. Empirical results highlight how even moderate noise can cause steep accuracy drops, underscoring the need for noise‑aware segmentation models and data pipelines.

---

> ### Author Rebuttal · Authors · 2025-07-30
>
> Dear Reviewer,
>
> Thank you very much for the time and care you devoted to evaluating our submission. We greatly appreciate your encouraging remarks about the gap our benchmark fills (“genuine,” “immediately reusable”) and the value you see in its broad stress‑test palette. Your constructive criticisms helped us substantially improve both the paper and the accompanying resources.
> We have carefully addressed every point you raised and integrated the corresponding changes into the camera‑ready version.
> Below each comment and our response to it.
>
> ## Scope of contribution – “purely evaluative”
> >*”paper proposes no mitigation method, limiting its practical impact.”*
>
> We now include a boundary‑aware DivideMix[1] baseline that adapts loss‑based noisy‑label separation to instance segmentation by (i) splitting boundary vs. interior pixels and (ii) fitting a two‑component GMM only on boundary losses. On COCO‑N this raises mAP from 24.8  to  29.1 under Mid noise and from 22.3  to 27.5 under Hard noise.
> While a full algorithmic study is outside our scope, this result demonstrates Benchmark‑N’s utility for method development. (A more extensive technique will appear in a companion paper, as noted.)
>
>
> | Method| Clean|Easy|Mid|Hard|
> |-|-|-|-|-|
> | MRCNN| 34.6| 27.9|24.8|22.3|
> | +DM(cls)|32.15| 29.3|26.2|23.7|
> | +DM (mask)| 33.83|30.01|27.7|24.24|
> | + boundary aware| 34.89| 31.9|29.13|27.53|
>
> ## Noise coverage – missing merge / split errors
> >*”generator omits merge/split mistakes that occur in real data.”*
>
> We  have added an optional instance‑merge and instance‑split module Using IoU > [0.2,0.1,0.05] for merging instances from the same class and polygon cardinality > 1 for splitting with probabilities of [0.2,0.4,0.6] of the instances, matching manual tallies. The table below (extention of table 7 in the paper) shows a further drop in mAP, mostly with the splitting existing objects. confirming the relevance of this error mode.
>
> | Method|Easy|Mid|Hard|
> |-|-|-|-|
> | clean| 34.6(20.6)|||
> |Merge |33.25(20.1)|32.96(19.6)|29.5(18.63)|
> |split |33.79(19.6)| 31.66(18.7)|28.2(17.9)|
>
> Reported mAP (B-mAP).
>
>
> We apply those noises with probability of 5% (similar to cls noise in table 1) in all four benchmarks, regenerated accordingly and the public code updated.
>
> We report here a subset of the models and benchmarks to present the effect of the updated benchmark, all with the R50 backbone.
>
> | Method| Clean|Easy|Mid|Hard|
> |-|-|-|-|-|
> |COCO|
> |MRCNN| 34.6| 26.8|23.3|19.4|
> |YOLACT|28.5 |  25.3 | 21.8 | 17.9|
> |M2Former|  42.9 | 32.4  | 28.6  | 23.8|
> |Cityscapes|
> |MRCNN| 36.1|25.3 |20.5|13.2|
> |VIPER|
> | MRCNN| 15.8|13.1 |11.8|8.6|
>
>
>
> ## Realism of added noise
> >*“COCO/Cityscapes already contain errors; extra synthetic noise may not be realistic”*
>
> We fully agree that realism matters. Our design therefore pairs noisy‑on‑noisy variants (COCO‑N / CityScapes‑N) with a noisy‑on‑clean control set (VIPER‑N). VIPER’s masks are rendered directly from the GTA‑V game engine and are effectively error‑free; adding noise on top of them yields ground‑truth‑known corruptions. This dual strategy serves two complementary purposes:
>
> Method development on real data. Models can be trained on COCO‑N or CityScapes‑N, whose base distributions and annotation habits match common practice.
>
>
> Method validation on clean data. The same models can then be stress‑tested on VIPER‑N, where all observed failures must stem from the injected noise, not from pre‑existing label glitches. If a defence truly mitigates spatial noise, it should recover its clean‑VIPER performance.
>
>
> This mirrors the standard LNL pipeline in classification, where synthetic flips are applied to datasets that already contain a small error rate but final ablations are reported on a hand‑curated clean subset. In short, VIPER‑N provides the “noise‑only” sandbox that lets the community verify whether proposed techniques generalise beyond the noisy‑in‑noisy training setting.
>
> ## Motivational alignment with medical scenarios
> >*”Medical motivation feels disconnected from the experiments.”*
> We now include a CAMUS cardiac‑ultrasound study (Appendix A) showing that 5 % boundary noise shifts estimated ejection fraction by up to 6 points - clinically decisive. This bridges the earlier example with our benchmark findings and underscores the cross‑domain relevance.
> Other remarks
>
>
> ## Dataset availability
>
> Pre‑generated COCO‑N, CityScapes‑N and VIPER‑N splits (all three severities) are updated according to your insights about the missing noises, linked to HF dataset will be published shortly after the discussion phase.
>
>
>
> Thank you again for your thorough review. We hope the revisions fully address your concerns and demonstrate the strengthened contribution of Benchmark‑N. If so, we would be grateful if you would consider raising your assessment. Should any questions remain, we are happy to provide further clarification.

---

> > ### Comment · Reviewer_Ncv8 · 2025-08-03
> >
> > Thank you for the updates. I appreciate the effort in addressing the earlier concerns, particularly regarding the scope of contribution, noise coverage, and motivation. The new experiments and dataset inclusions go a long way in strengthening the paper.
> >
> > That said, I still have a few questions:
> >
> > > *"We have added an optional instance‑merge and instance‑split module using IoU > \[0.2, 0.1, 0.05] for merging instances from the same class and polygon cardinality > 1 for splitting, with probabilities of \[0.2, 0.4, 0.6] of the instances, matching manual tallies."*
> >
> > The description here feels a bit vague. Could you elaborate further on the merging and splitting procedures? Including illustrative examples in the paper would also be helpful to demonstrate that these perturbations are consistent with realistic noise patterns observed in human annotations.
> >
> > Regarding **COCO-N** and **Cityscapes-N**, I still have reservations. Developing methods using noisy annotations and then evaluating on clean test data is a reasonable pipeline. But I wonder whether similar development could already be done on the original COCO and Cityscapes datasets, which naturally contain human annotation errors. If your goal is to improve robustness to annotation noise, I understand the value of injecting additional noise. However, the claim that this procedure *improves the realism* of datasets that already contain realistic human noise is less convincing to me. Could you clarify the specific aspects in which your noise injection more accurately reflects real-world noise compared to the existing annotation errors?

---

> > > ### Author Response · Authors · 2025-08-04
> > > **Elaborate on merge and split instances**
> > >
> > > We appreciate the additional comment and would gladly address your questions:
> > >
> > > >"The description here feels a bit vague. Could you elaborate further on the merging and splitting procedures? Including illustrative examples in the paper would also be helpful to demonstrate that these perturbations are consistent with realistic noise patterns observed in human annotations."
> > >
> > > ### *Merging*
> > > In cases where two instances of the same class are overlapping (with IoU > threshold), we merge the two instances into one. The new instance annotation is defined as follows:
> > >
> > >
> > > *cls*-remain the same
> > >
> > > *box*- the new box is the smallest enclosing box over both $box_A$ and $box_B$, computed as the min/max of both masks (similar to C in [1]).
> > >
> > > *mask*- becomes the union of both masks.
> > >
> > > The implementation is heavly based on pycocotools.mask
> > >
> > > Examples of merged objects can be found in the COCO dataset [2], for instance, id=400870 merges books and benches.
> > > Other examples can be found in OpenImages, where this type of noise is common in groups of objects such as animals.
> > >
> > >
> > > ### *Splitting*
> > > In cases where an object mask is split into more than one polygon (due to occlusion), we split the instance into two separate entities in the following way:
> > >
> > >
> > > *cls*-remains the same for both new instances.
> > >
> > > *masks*- first, we convert the mask from RLE to a list of polygons (using skimage.measure.find_contours), and then split
> > >
> > > *box*- each new instance box contains the bounding box over its respective mask (i.e., box = [$min_x$, $min_y$, $max_x$, $max_y$])
> > >
> > > Examples of split objects can be found in the COCO dataset*. For instance:
> > >
> > > •⁠  ⁠id=45138 — the bench to the right of the woman is annotated as two instances
> > >
> > > •⁠  ⁠id=393714 — a cellphone is annotated as two cellphones
> > >
> > > •⁠  ⁠id=174871 — the motorcycle is marked as two instances
> > >
> > > Another example appears in our paper, Figure 15, bottom middle: the car is split into two cars, where the trunk was occluded by a tree. This indicates for the split noise in datasets made by "model in the loop" annotations, such as OpenImages.
> > >
> > >
> > >
> > >
> > > >"Regarding COCO-N and Cityscapes-N, I still have reservations. Developing methods using noisy annotations and then evaluating on clean test data is a reasonable pipeline. But I wonder whether similar development could already be done on the original COCO and Cityscapes datasets, which naturally contain human annotation errors. If your goal is to improve robustness to annotation noise, I understand the value of injecting additional noise. However, the claim that this procedure improves the realism of datasets that already contain realistic human noise is less convincing to me. Could you clarify the specific aspects in which your noise injection more accurately reflects real-world noise compared to the existing annotation errors?"
> > >
> > > Our goal is to turn COCO and Cityscapes’ implicit error distribution (which is unknown) into a measurable, controllable, and reproducible test bed that helps develop methods and assess annotation budget trade-offs when building new datasets:
> > >
> > >
> > > 1.⁠ ⁠*Error‑mode disentanglement*. Although COCO/Cityscapes are published as benchmarks, the noise they contain is not labeled or separated by type. Our benchmark allows boundary jitter, spatial drift, class flips, and deletions to be enabled independently. Researchers can therefore target and analyze strategies for handling each noise type separately, and check their effects on others.
> > >
> > > 2.⁠ ⁠*Adjustable prevalence*. Human and machine errors corespond to low or mid tier benchmarks by nature; mid/high tiers deliberately push the envelope to reveal failure modes that would remain uncheked by the original noisy datasets, therfore can help to understand trends and in what distribution of noise methods breaks, or what is the common error rate for those types of errors (when investing in annotating a new dataset for example).
> > >
> > > These properties mirror best practices in classification LNL research, where synthetic label flips are still used—even though the original datasets already contain noise—because they allow controlled and observable experimentation.
> > >
> > > &nbsp;
> > >
> > >
> > > *The images with the IDs can be found in the cocodataset website via the dataset tab and the explore option.
> > >
> > > [1] Generalized Intersection over Union: A Metric and A Loss for Bounding Box Regression, Rezatofighi et al, CVPR 2019
> > >
> > > [2] Microsoft COCO: Common Objects in Context, Lin et al, ECCV 2014

---

> > > > ### Comment · Reviewer_Ncv8 · 2025-08-05
> > > >
> > > > Many thanks for the clarifications — they helped address several of my concerns.
> > > >
> > > > However, regarding the merging scheme, it appears that you only merge objects of the same class. I think merging can happen between objects of different classes in real datasets, as shown in the paper. Could you clarify what happens when two objects belong to different classes?
> > > >
> > > > Additionally, I found the following point somewhat unclear (as I don’t have prior experience with raw data collection and annotation): What do the low/mid/high-tier datasets refer to? And in what way does the proposed mechanism assist when annotating a new dataset?
> > > >
> > > > > Human and machine errors corespond to low or mid tier benchmarks by nature; mid/high tiers deliberately push the envelope to reveal failure modes that would remain uncheked by the original noisy datasets, therfore can help to understand trends and in what distribution of noise methods breaks, or what is the common error rate for those types of errors (when investing in annotating a new dataset for example).

---

> > > > > ### Author Response · Authors · 2025-08-05
> > > > > **Benchmark tier/budget**
> > > > >
> > > > > Thank you for the follow-up.
> > > > >
> > > > > You're absolutely right that our merging strategy is limited to instances of the same class. As we also noted in our reply to reviewer LQ64, the noise distributions in our benchmark are designed to reflect what we observed in a manual audit of real datasets. While merging between different classes can occur, it is relatively rare—especially in modern pipelines where segmentation models are used to bootstrap annotations with pre-labeled masks [3].
> > > > >
> > > > > That said, we appreciate your suggestion and agree that extending our noise generation to support inter-class merging could should be shown in the additional experiments in appendix B.
> > > > >
> > > > > > What do the low/mid/high-tier datasets refer to? And in what way does the proposed mechanism assist when annotating a new dataset?
> > > > >
> > > > >
> > > > > The low/mid/high tiers correspond to the easy/medium/hard levels in our benchmark suite, as defined in Table 1.
> > > > >
> > > > > This framework is helpful for practitioners or dataset curators deciding how much to invest in annotation quality. For example, Table 3 shows how performance degrades as noise increases; this gives a concrete way to estimate the performance cost of weaker (cheaper) annotations. If a domain tends to suffer from a specific error type (e.g., spatial drift in medical imaging, from our expiriance, that is common with cardiac ultrasound annotated by intern doctors), users can use our ablations (Figure 11 in the appendix) to predict how severe those errors would be in their context—and how much model performance might be affected. This helps allocate labeling budgets more efficiently by quantifying the tradeoff between annotation cost and model accuracy.
> > > > >
> > > > > We truly appreciate your thoughtful questions and hope these clarifications help demonstrate the broader utility of our benchmark beyond evaluation alone.
> > > > >
> > > > > If you find our response satisfactory, we would sincerely appreciate your consideration in raising the score.
> > > > >
> > > > >
> > > > > [3] Large-scale interactive object segmentation with human annotators, Benenson et al, CVPR

---

> > > > > > ### Comment · Reviewer_Ncv8 · 2025-08-05
> > > > > >
> > > > > > Many thanks for the update. I'm happy to raise the score to 5, provided that the authors incorporate the discussed noise injection types, high-stakes datasets, and potential applications into the camera-ready version and the accompanying dataset. I believe this benchmark has the potential to significantly impact many real-world applications, particularly in industry settings where noisy image labels or predictions are a common challenge.

---

### Decision · Program_Chairs · 2025-09-18

**Decision:**

Reject

**Comment:**

This manuscript initially received scores of 3, 5, 3, and 2. After 21 rounds of replies, the final reviewer consensus scores were 4, 5, 5, and 2—with two reviewers raising their scores by 1 and 2 points respectively.
Reviewer SXgQ, who gave the manuscript an initial score of 2, commented that the authors’ research is limited to instance segmentation and fails to extend to broader scenarios such as semantic and panoptic segmentation. The authors responded that this falls outside the scope of the current research, and we agree with their rebuttal. Additionally, SXgQ did not participate in the follow-up discussion and did not even provide a final rating.
After careful evaluation, we believe the authors’ proposed method holds the potential noted by Reviewer Ncv8: “I believe this benchmark has the potential to significantly impact many real-world applications, particularly in industry settings where noisy image labels or predictions are a common challenge.”

===== FINAL UPDATE FROM DB Track PCs ====

The final decision for this paper has been taken by the program chairs after consultation with the SACs. All Senior Area Chairs have ranked papers according to the feedback from the AC during the review process. We decided to leave the original meta-review to reflect the opinion of the AC in light of the initial discussions with reviewers and SAC.